# A Deep Dive into Dataset Imbalance and Bias in Face Identification

## Abstract

As the deployment of automated face recognition (FR) systems proliferates, bias in these systems is not just an academic question, but a matter of public concern. Media portrayals often center imbalance as the main source of bias, i.e., that FR models perform worse on images of non-white people or women because these demographic groups are underrepresented in training data. Recent academic research paints a more nuanced picture of this relationship. However, previous studies of data imbalance in FR have focused exclusively on the face *verification* setting, while the face *identification* setting has been largely ignored, despite being deployed in sensitive applications such as law enforcement. This is an unfortunate omission, as 'imbalance' is a more complex matter in identification; imbalance may arise in not only the training data, but also the testing data, and furthermore may affect the proportion of identities belonging to each demographic group *or* the number of images belonging to each identity. In this work, we address this gap in the research by thoroughly exploring the effects of each kind of imbalance possible in face identification, and discuss other factors which may impact bias in this setting.

## 1 Introduction

Automated face recognition is becoming increasingly prevalent in modern life, with applications ranging from improving user experience (such as automatic face-tagging of photos) to security (e.g., phone unlocking or crime suspect identification). While these advances are impressive achievements, decades of research have demonstrated disparate performance in FR systems depending on a subject's race (Phillips et al., 2011; Cavazos et al., 2020), gender presentation (Alvi et al., 2018; Albiero et al., 2020), age (Klare et al., 2012), and other factors. This is especially concerning for FR systems deployed in sensitive applications like law enforcement; incorrectly tagging a personal photo may be a mild inconvenience, but incorrectly identifying the subject of a surveillance image could have life-changing consequences. Accordingly, media and public scrutiny of bias in these systems has increased, in some cases resulting in policy changes.

One major source of model bias is dataset imbalance; disparities in rates of representation of different groups in the dataset. Modern FR systems employ neural networks trained on large datasets, so naturally much contemporary work focuses on what aspects of the training data may contribute to unequal performance across demographic groups. Some potential sources that have been studied include imbalance of the proportion of data belonging to each group (Wang & Deng, 2020; Gwilliam et al., 2021), low-quality or poorly annotated images (Dooley et al., 2021), and confounding variables entangled with group membership (Klare et al., 2012; Kortylewski et al., 2018; Albiero et al., 2020).

Dataset imbalance is a much more complex and nuanced issue than it may seem at first blush. While a naive conception of 'dataset imbalance' is simply as a disparity in the *number* of images per group, this disparity can manifest itself as either a gap in the number of identities per group, or in the number of images per identity. Furthermore, dataset imbalance can be present in different ways in both the training and testing data, and these two source of imbalance can have radically different (and often opposite) effects on downstream model bias.

Past work has only considered the *verification* setting of FR, where testing consists of determining whether a pair of images belongs to the same identity. As such, 'imbalance' between demographic groups is not a meaningful concept in the test data. Furthermore, the distinction between imbalance

Figure 1: **Examples of imbalance in face identification**. Top left: data containing more female identities than male identities. Top right: data containing the same number of male and female identities, but more images per male identity. Bottom: two possible test (gallery) sets showing how the effects of different kinds of imbalance may interact.

of identities belonging to a certain demographic group versus that of images per identity in each demographic group has not been carefully studied in either the testing or the training data. All of these facets of imbalance are present in the face *identification* setting, where testing involves matching a probe image to a gallery of many identities, each of which contains multiple images. We illustrate this in Figure 1.

In this work, we unravel the complex effects that dataset imbalance can have on model bias for face identification systems. We separately consider imbalance (both in terms of identities or images per identity) in the train set and in the test set. We also consider the realistic social use case in which a large dataset is collected from an imbalanced population and then split at random, resulting in similar dataset imbalance in both the train and test set. We specifically focus on imbalance with respect to gender presentation, as (when restricting to only male- and female-identified individuals) this allows the proportion of data in each group to be tuned as a single parameter, as well as the availability of an ethically obtained identification dataset with gender presentation metadata of sufficient size to allow for subsampling without significantly degrading overall performance.

Our findings show that each type of imbalance has a distinct effect on a model's performance on each gender presentation. Furthermore, in the realistic scenario where the train and test set are similarly imbalanced, the train and test imbalance have the potential to interact in a way that leads to systematic underestimation of the true bias of a model during an audit. Thus any audit of model bias in face identification must carefully control for these effects.

The remainder of this paper is structured as follows: Section 2 discusses related work, and Section 3 introduces the problem and experimental setup. Sections 4 and 5 give experimental results related to imbalance in the training set and test set, respectively, and Section 6 gives results for experiments where the imbalance in the training set and test set are identical. In Section 7.1, we evaluate randomly initialized feature extractors on test sets with various levels of imbalance to further isolate the effects of this imbalance from the effects of training. In Section 7.2, we investigate the correlation between the performance of models trained with various levels of imbalance and human performance.

## 2 RELATED WORK

### 2.1 IMBALANCE IN VERIFICATION

Even before the advent of neural network-based face recognition systems, researchers have studied how the composition of training data affects verification performance. Phillips et al. (2011) compared algorithms from the Face Recognition Vendor Test (Phillips et al., 2009) and found that those developed in East Asia performed better on East Asian Faces, and those developed in Western

countries performed better on Caucasian faces. Klare et al. (2012) expanded on these results by comparing performance across race, gender presentation, and age cohorts, observing that training exclusively on images of one demographic group improved performance on that group and decreased performance on the others. They further conclude that training on data that is "well distributed across all demographics" helps prevent extreme bias.

Multiple verification datasets have been proposed in the interest of eliminating imbalance as a source of bias in face verification. The *BUPT-BalancedFace* dataset (Wang & Deng, 2020) contains an approximately equal number of identities and images of four racial groups[1]. *Balanced Faces in the Wild* (Robinson et al., 2020) goes a step further, balancing identities and images across eight categories of race-gender presentation combinations. Also of note is the *BUPT-CBFace* dataset (Zhang & Deng, 2020), which is class-balanced (each identity possesses the same number of images), rather than demographically balanced.

Some recent work in verification has questioned whether perfectly balanced training data is in fact an optimal setting for reducing bias. Albiero et al. (2020) studied sources of bias along gender presentation; among their findings, they observe that balancing the amount of male and female training images and identities in the training data reduces, but does not eliminate, the performance gap between gender presentations. Similarly, Gwilliam et al. (2021) trained models on data with different racial makeups, finding that models which were trained with more images of African subjects had lower variance in performance on each race than those which were trained on balanced data.

## 2.2 Bias in Identification

Although the effect of imbalance on bias has only been explicitly studied in face verification, there is some research on identification which is relevant. The National Institutes of Standards and Technology performed large-scale testing of commercial identification algorithms, finding that many (though not all) exhibit gender presentation or racial bias (Grother et al., 2019). The evaluators speculate that the training data or procedures contribute to this bias, but could not study this hypothesis due to the proprietary nature of the models. Dooley et al. (2021) evaluated commercial and academic models on a variant of identification in which each probe image is compared to 9 gallery images of distinct identities, but belonging to the same skin type and gender presentation. They find that academic models (and some, but not all, commercial models) exhibit skin type and gender presentation bias despite a testing regime which makes imbalance effectively irrelevant.

## 2.3 Imbalance in Deep Learning

Outside the realm of facial recognition, there is much study about the impacts of class imbalance in deep learning. In standard machine learning techniques, i.e., non-deep learning, there are many well-studied and proven techniques for handling class imbalances like data-level techniques (Van Hulse et al., 2007; Chawla et al., 2002; 2004), algorithm-level methods (Elkan, 2001; Ling & Sheng, 2008; Krawczyk, 2016), and hybrid approaches (Chawla et al., 2003; Sun et al., 2007; Liu et al., 2008). In deep learning, some take the approach of random over or under sampling (Hensman & Masko, 2015; Lee et al., 2016; Pouyanfar et al., 2018). Other methods adjust the learning procedure by changing the loss function (Wang et al., 2016) or learning cost-sensitive functions for imbalanced data (Khan et al., 2017). We refer the reader to Buda et al. (2018); Johnson & Khoshgoftaar (2019), for a thorough review of deep learning-based imbalance literature. Much of the class-imbalance work has been on computer vision tasks, though generally has not examined specific analyses like we present in this work like network initialization, face identification, or intersectional demographic imbalances.

## 2.4 Other sources of bias in facial recognition

Face recognition is a complex, sociotechnical system where biases can originate from the algorithms (Danks & London, 2017), preprocessing steps (Dooley et al., 2020), and human interpretations (Chouldechova & Roth, 2020). While we do not explicitly examine these sources, we refer the reader to Mehrabi et al. (2021); Suresh & Guttag (2019) for a broader overview of sources of bias in machine learning.

---

[1]This work also introduces *BUPT-GlobalFace*, which instead approximately matches the distribution across races to that of the world population.

## 3    FACE IDENTIFICATION SETUP

Face recognition has two tasks: face verification and face identification. The first refers to verifying whether a person of interest (called the *probe image*) and a person in a reference photo are the same. This is the setting that might be applied, e.g., to phone unlocking or other identity confirmation. In contrast, face identification involves matching a probe image against a set of images (called the *gallery*) with known identities. This application is relevant to search tasks, such as identifying the subject of a photo from a database of driver's license or mugshot photos.

In a standard face recognition pipeline, an image is generally first pre-processed by a face detection system which may serve to locate and align target faces to provide more standardized images to the recognition model. State-of-the-art face recognition models exploit deep neural networks which are trained on large-scale face datasets for a classification task. At test time, the models work as feature extractors, so that the similarity between a probe image and reference photo (in verification) or gallery photos (in identification) is computed in the feature space. In verification, the similarity score is then compared with a predefined threshold, while in identification a k-nearest neighbors search is performed using the similarity scores with the gallery images.

We focus on the face identification task in our experiments and explore how different kinds of data balance affect the models performance across demographic groups (specifically, the disparity in performance on male and female targets). We also analyze how algorithmic bias correlates with human bias on InterRace, a manually curated dataset specifically designed for bias auditing, with challenging face recognition questions and provided annotations for gender presentation and skin color Dooley et al. (2021).

Our experiments use state-of-the-art face recognition models. We train MobileFaceNet (Chen et al., 2018), ResNet-50, and ResNet-152 (He et al., 2016) feature extractors each with a CosFace and ArcFace head which improve the class separability of the features by adding angular margin during training (Deng et al., 2019; Wang et al., 2018). For training and evaluation we use the CelebA dataset (Liu et al., 2015), which provides annotations for gender presentation. As our main research questions focus on the impact of class imbalance, we pay special attention to the balance of the gender presentation attribute in our training. The original dataset contains more female identities. As such, we create a balanced training set containing 140,000 images from 7,934 identities with equal number of identities and total number of images from each gender presentation. We also create a perfectly balanced test set containing 14,000 images from 812 identities. The identities in the train and test sets are disjoint. We call these the *default train* and *default test* sets. All models are trained with class-balanced sampling to ensure equal contribution of identities to the loss. We additionally include results for models trained without over-sampling in Appendix A.5.

Recall that our research question is to investigate how class imbalances affect face identification. In order to answer this question, we train models on a range of deliberately imbalanced subsamples of the default training set, and test models on a range of deliberately imbalanced subsamples of the default test set, in order to explore the impact on the model's performance for each gender presentation.

To evaluate the models, we compute rank-1 accuracy over the test set. Specifically, for each test image we treat the rest of the test set as gallery images and find if the closest gallery image in the feature space (as defined by cosine similarity) of a model is an image of the same person.

When we make comparisons with human performance (Section 7.2), we use the InterRace dataset Dooley et al. (2021). Since the InterRace dataset is derived from both the CelebA and LFW (Huang et al., 2007) datasets, we additionally train models on the InterRace-train split of CelebA, containing images of identities not included in the InterRace dataset. Similar to other experiments, we train models with varying levels of either identity and image imbalance.

## 4    BALANCE IN THE TRAIN SET

### 4.1    BALANCING THE NUMBER OF IDENTITIES

**Experiment Description.** To explore the effect of train set balance in the number of identities on gender presentation bias, we construct train data splits with different ratios of female and male identities, while ensuring that the average number of images per identity is the same across gender

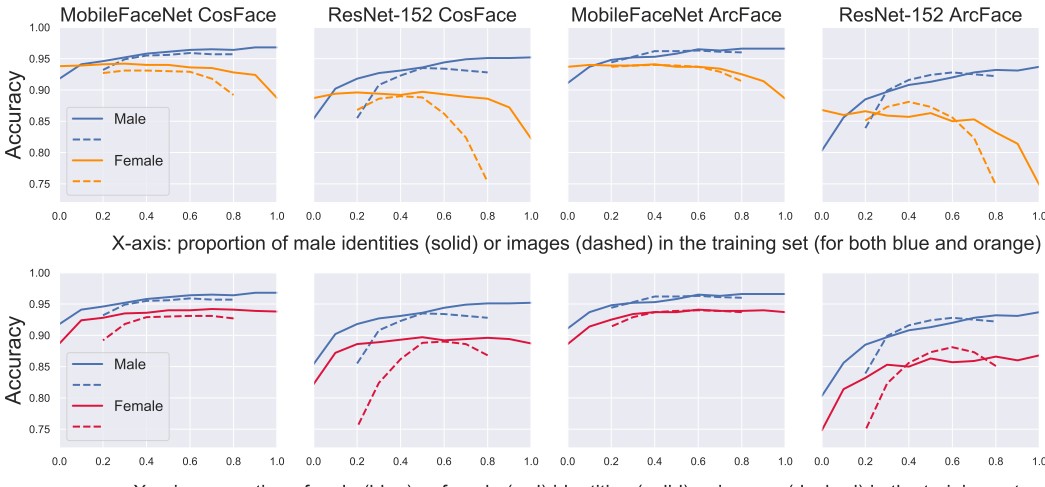

Figure 2: **Train Set Imbalance.** Results of experiments that change the train set gender presentation balance. Top row: male and female accuracy are plotted against the proportion of male data in the train set. Bottom row: for an alternate view, female accuracy is flipped horizontally, so that it is plotted against the proportion of *female* data in the train set. All models are tested on the default balanced test set.

presentations. Therefore, in all splits we have the same total number of images and total number of identities, but the proportion of female and male identities varies. We consider splits with $0:10$, $1:9, 2:8, ..., 10:0$ ratios, each having 70,000 total images from 3967 identities. We evaluate the models on the (perfectly balanced) default test set and report rank-1 face identification accuracy as described in Section 3. More details of train set splits can be found in Table 1.

**Results.** We compute accuracy scores separately for male and female test images for models trained on each of the train splits and depict them in Figure 2 with solid lines. From the first row plots, we observe that a higher proportion of male identities in the train set leads to an increase in male accuracy and decrease in female accuracy, with the most significant drops occurring near the extreme $10:0$ imbalance. This indicates that it is very important to have at least a few identities from the target demographic group in the train set; once the representation of the minority group reaches 10%, the marginal gain of additional identities becomes less. We also observe that for most models, the female accuracy drops slightly when the proportion of female identities exceeds 80% of the training data, which does not happen to the male group. Consult Table 2 for the numerical results.

Regarding the model architectures, MobileFaceNet models trained with both CosFace and ArcFace heads outperform ResNet models on both female and male images and have smaller absolute accuracy gap. However, the error ratio is similar across the models, see Table 2. Finally, the accuracy gap is closed for all models when the train set consists of about 10% male and 90% female identities.

In addition, in the second row of Figure 2 we compare how similar these trends are for females and males by plotting female accuracy against the proportion of *female* identities in the train set. One can see that for MobileFaceNet models the accuracy on male and female images increases similarly when increasing the proportion of "target" identities up to 80%. However, for ResNet models adding more female identities in the train set results in smaller gains compared to the effect of adding more male identities on male accuracy.

### 4.2 BALANCING THE NUMBER OF IMAGES PER IDENTITY

In the previous subsection, we fixed the average number of images per identity in each gender presentation and adjusted the number of identities. We now will do the reverse: fix the number of identities and vary the images per identity.

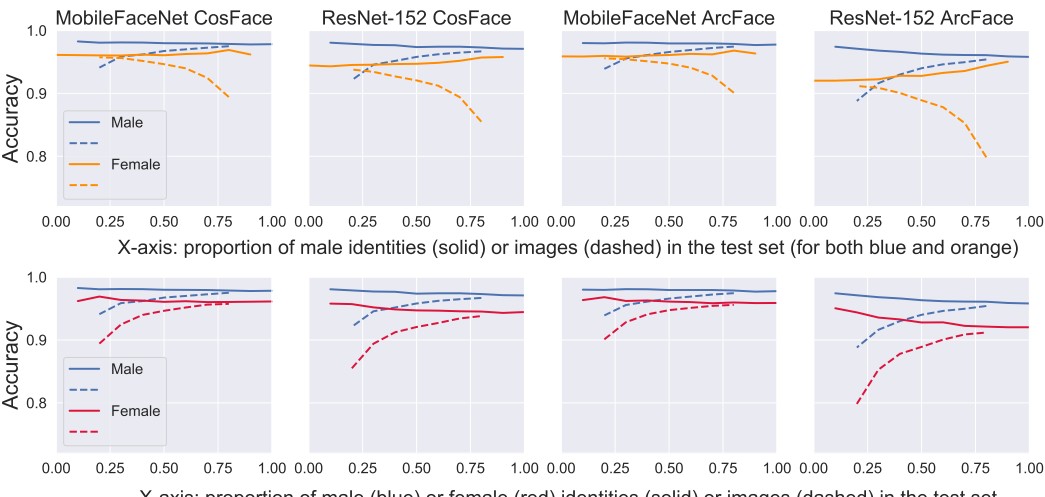

Figure 3: **Test Set Imbalance.** Results of experiments that change the test set gender presentation balance. Top row: male and female accuracy are plotted against the proportion of male data in the test set. Bottom row: for an alternate view, female accuracy is flipped horizontally, so that it is plotted against the proportion of *female* data in the test set. All models are trained on the default balanced train set. For each experiment, the test set was split with 5 random seeds, and the results are averaged across seeds.

**Experiment Description.** We change the average number of images per male and female identity, but fix the number of identities of each gender presentation. We consider ratios $2 : 8, ..., 8 : 2$, each having $70,000$ images from $7,934$ identities. We do not consider more extreme ratios, which would result in identities with fewer than 3 images.

**Results.** The dashed lines in Figure 2 illustrate the accuracy of the models trained on described data splits. From the first row plots we see that, similar to the previous experiment, increasing the number of male images in the train set leads to increased accuracy on male and decreased accuracy on female images. Interestingly, we observe a decrease in performance for both demographic groups when the images of that group constitute more than $60\%$ of train data; this is most easily visible in the second row of Figure 2. However, we find that this effect results from the widely used class-balanced sampling training strategy, and models trained without the default oversampling are more robust to imbalance in the number of images per identity, see details in Section A.5 and Figure 8. The "fair point" where female accuracy is closest to male accuracy occurs when around $20\%$ of images are of males.

When comparing the effect of imbalance in the number of identities and the number of images per identity (solid and dashed lines respectively in Figure 2), we see that ResNet models are more susceptible to image imbalance than to identity imbalance, which is also a phenomenon specific to the common class-balanced sampling.

## 5 BALANCE IN THE TEST SET

### 5.1 BALANCING THE NUMBER OF IDENTITIES

**Experiment Description.** Analogous to the train set experiments, we split the test data (the gallery) with different ratios of female and male identities, while keeping the same average number of images per identity for both demographic groups. For each ratio, we split the test data with 5 random seeds and report average rank-1 accuracy of the models trained on default train data. The results are shown in the solid lines of Figure 3, as well as in Table 4.

**Results.** We observe that increasing the proportion of identities of a target demographic group in the test set hurts the model's performance on that demographic group, and this trend is similar for male and female images. Intuitively, this is because face recognition models rarely match images to one

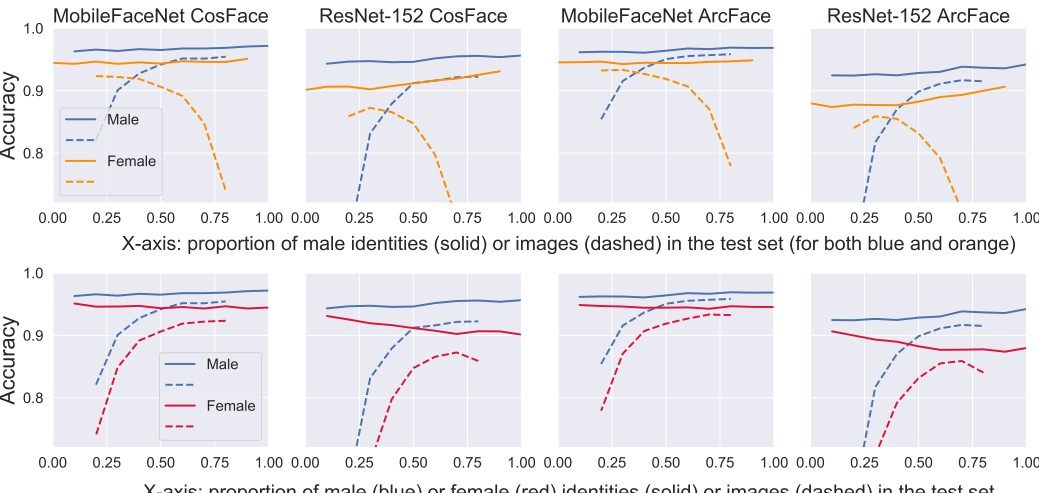

Figure 4: **Train & Test Set Imbalance.** Results of experiments that adjust the gender presentation balance in both the train and test set. Top row: male and female accuracy are plotted against the proportion of male data used in both the train and test set. Bottom row: for an alternate view, female accuracy is flipped horizontally, so that it is plotted against the proportion of *female* data in both the train and test set. For each experiment, the test set was split with 5 random seeds, and the results are averaged across seeds.

of a different demographic group; therefore by adding more identities of a particular demographic group, we add more potential false matches for images from that demographic group, which leads to higher error rates. We also see that ResNet models are more sensitive to the number of identities in the gallery set than MobileFaceNet models.

## 5.2 Balancing the number of images per identity

**Experiment Description.** Now, we investigate how increasing or decreasing the number of images per identity affects the performance and bias of the models. Again, we split the test sets with different ratios of total number of images across gender presentations, but same number of identities, each with 5 random seeds. These results are recorded as dashed lines in Figure 3, as well as in Table 5.

**Results.** Unlike the results with identity balance, increasing the average number of images per identity leads to performance gains, since this increases the probability of a match with an image of the same person. Also, image balance affects the performance more significantly than identity balance, and these trends are similar across all the models and both gender presentations. Finally, we note that the "fair point" for image balance in the test set occurs at about 30% male images; contrast this with identity balance, for which no fair point appears to exist.

## 6 A cautionary tale: matching the balance in the train and gallery data

Using our findings from above, we conclude that common machine learning techniques to create train and test splits can lead to Simpson's paradoxes which lead to a false belief that a model is unbiased. It is standard practice to make random train/test splits of a dataset. If the original dataset is imbalanced, as is commonly the case, the resulting splits will be imbalanced in similar ways. As we have seen above, the effects of imbalance in the train and test splits may oppose one another, causing severe underestimation of model bias when measured using the test split. This occurs because the minority status of a group in the train split will bias the model towards low accuracy on that group, while the correspondingly small representation in the test split will cause an increase in model accuracy, partially or entirely masking the true model bias. The results for these experiments are presented in Figure 4 and Tables 6, 7.

**Balancing the number of identities** We create train and test sets with identical distributions of identities. Recalling the results from prior experiments, increasing the number of identities for the target group in the training stage improves accuracy on that group, while adding more identities in the gallery degrades it. Interestingly, when we increase the proportion of male identities in *both* train and test sets, we observe gains in both male and female accuracy, and that trend is especially strong for ResNet models.

**Balancing the number of images per identity** Having more images is beneficial in both train and test stages. Therefore, the effect of image balance is amplified when both train and test sets are imbalanced in a similar way. Similar to the train set experiments, having more than 70% female images in both train and test sets leads to slight drops in female accuracy on ResNet models, which again is a result of the default class-balanced oversampling strategy.

## 7 BIAS COMPARISONS

We ask two concluding questions: one about whether class imbalance captures all the inherent bias and the other about how the bias we see compares to human biases. First, we explore how data imbalances cause biases in random networks and find surprising conclusions. Then, we ask how class imbalances in machines compare to how humans exhibit bias on face identification tasks.

### 7.1 BIAS IN RANDOM FEATURE EXTRACTORS

Given a network with random initializations, we would expect that evaluation on a balanced test set would result in equal performance on males and females, and likewise that male performance on a set with a particular proportion of male identities would be the same as female performance when that proportion is reversed. However, this is not the case. We test randomly initialized feature extractors on galleries with varying levels of image imbalance. Figure 5 summarizes the results of these experiments. We observe that both models have higher male performance when the test set is perfectly balanced, and that performance on males is higher when they make up 80% of the test set than female performance when they make up 80% of the test set. This provides strong evidence that there are sources of bias that lie outside what we explore here and which are potential confounders to a thorough study of bias in face identification; further work on this is warranted.

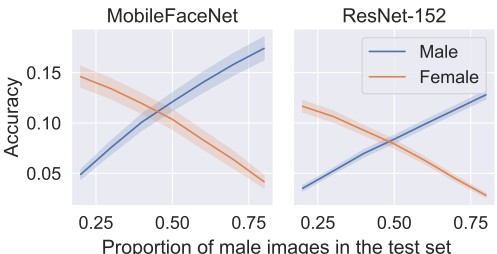

Figure 5: **Random Feature Extractors.** The plot illustrates male (blue) and female (orange) accuracy of random feature extractors against the proportion of male images in the test set. The standard deviation is computed across 10 random initializations.

### 7.2 ARE MODELS BIASED LIKE HUMANS?

Numerous psychological and sociological studies have identified gender, racial, and other biases in human performance on face recognition tasks. Dooley et al. (2021) studied whether humans and FR models exhibit similar biases. They evaluated human and machine performance on the curated InterRace test questions, and found models indeed tend to perform better on the same groups as, and with comparable gender presentation bias ratios to, humans. In this section, we use their human survey data to explore two related questions: how correlated are model and human performance *at the question level*, and how does this change with different levels of imbalance in training data?

To answer these questions, we define a metric which allows us to distinguish how well a model performs on each InterRace identification question. Let

$$L_2 \text{ ratio} = \frac{\|v_{probe} - v_{false}\|_2}{\|v_{probe} - v_{true}\|_2 + \|v_{probe} - v_{false}\|_2},$$

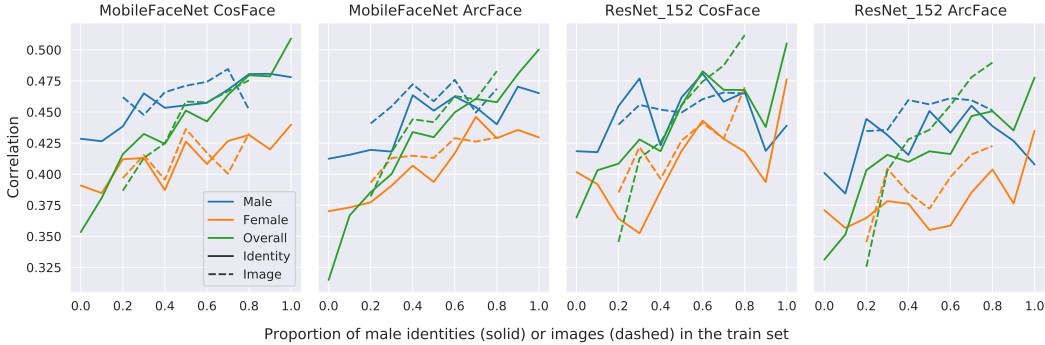

Figure 6: Pearson correlation of $L2$ ratio vs. human accuracy for various models as proportion of male training data varies.

where $v_{probe}$, $v_{true}$, $v_{false}$ are the feature representations of the probe image, the correct gallery image, and the nearest incorrect gallery image, respectively.[2] This value is 1 when the probe and correct image's representations coincide, 0 when the probe and incorrect image's representations coincide and 0.5 when the probe's representation is equidistant from those of the correct and incorrect image. Figure 7 depicts examples of scatterplots comparing model confidence to human accuracy on each InterRace question.

Figure 6 shows the correlation between $L2$ ratio and human performance for various models at each of the training imbalance settings that we have considered in earlier experiments. We see that the correlation between these values over *all* questions tends to rise as the proportion of male training data increases. However, the correlation when separately considering male and female questions does not rise as monotonically, or as much, from left to right as the overall correlation does. This suggests that the correlation between human and machine performance is largely driven by the fact that models and humans both find identifying females more difficult than identifying males, and that this disparity is exacerbated when the model in question is trained on male-dominated data. On the other hand, the *particular* males and females that are easier or harder to identify appear to differ between models and humans, which suggests the *reasons* for bias in humans and machines are different.

# 8   ACTIONABLE INSIGHTS

We note five actionable insights for machine learning engineers and other researchers from this work. First, **overrepresenting the target demographic group can sometimes hurt that group**. Sometimes having more balanced data is the key. Also, class-balanced sampling might hurt representation learning when the data is not balanced with respect to the number of images per identity. Second, **gallery set balance is as important as train set balance**, contrary to how face verification class imbalances work. Third, **having the same distribution of identities and average number of images per identity is not an unbiased way to evaluate a model**, since the effects of balance in train and test sets can be amplified (in case of images) or cancel each other (in case of identities). Fourth, **train and test class imbalances are not the only cause of bias** in face identification evaluation since even random models do not perform equally poorly on female and male images. Finally, even though both humans and machine find female images more difficult to recognize, **it seems that the reasons for bias are different in people and models**. We know that this work sheds light on common mistakes in bias computations for many facial recognition tasks and hope that auditors and engineers will incorporate our insights into their methods.

---

[2]We note that other measures of confidence in a $k$-nearest neighbors setting, such as those discussed in (Dalitz, 2009), are inappropriate for this application.

## 9 REPRODUCIBILITY STATEMENT

We include the code and instructions for reproducing the results of our experiments in the supplementary materials. In our experiments we use CelebA facial dataset, which is publicly available for research purposes only (Liu et al., 2015) and can be downloaded at https://mmlab.ie.cuhk.edu.hk/projects/CelebA.html. We also include details on hyperparameters used to train the models in Appendix A.2

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

# A APPENDIX

## A.1 BROADER IMPACT AND LIMITATIONS

In this work we explore the effects of various forms of data imbalance on bias in face identification, and we hope that practitioners will take our findings into account when performing bias auditing. However, it is important to understand that biases can originate from various sources besides data imbalance and therefore models should be carefully evaluated for other bias factors.

The availability of high quality datasets, which are suitable for the identification task (as opposed to verification), have demographic metadata/annotation for both train and test sets, and contain enough identities and images belonging to each demographic group to allow for subsampling, is extremely limited. For this reason we focus solely on gender bias and leverage CelebA dataset, which meets these criteria. Also, using a binary demographic attribute (such as gender, when restricting only to male- and female-presenting identities) allows the proportion of data in each group to be conveniently tuned as a single parameter, which in turn makes interpreting results more straightforward.

### A.1.1 DETAILS ON CELEBA

CelebA is a publicly available dataset, which is constructed from CelebFaces dataset Sun et al. (2014) and contains face images collected from the Internet.

## A.2 TRAINING DETAILS

We pre-process CelebA images by aligning them using the provided facial landmarks and cropping to 112x112 size. All face recognition models are trained with Focal loss Lin et al. (2017) using SGD for 100 epochs with learning rate of 0.1, momentum of 0.9 and weight decay of 5e-4. The learning rate is reduced by 10 times at epochs 35, 65 and 95. Horizontal flip data augmentation is used during training. For the model architectures, we use implementation from publicly available github repository `face.evoLVe.PyTorch`[3]. We run our experiments on NVIDIA GeForce RTX 2080 Ti machines and each experiment takes from 6 to 12 hours of compute time on one GPU.

## A.3 EXPERIMENTAL DETAILS

Table 1: Details on the number of identities, total number of images and average number of images per identity used in experiments with train and test data balance. We also report statistics for the default train and test sets. M denotes male, F denotes female.

| Setting | M ids | F ids | Total M imgs | Total F imgs | M imgs/id | F imgs/id | Total ids | Total imgs |
|---|---|---|---|---|---|---|---|---|
| Train default | 3967 | 3967 | 70k | 70k | 17.65 | 17.65 | 7934 | 140k |
| Train id balance | 0 - 3967 | 0 - 3967 | 0 - 70k | 0 - 70k | 17.65 | 17.65 | 3967 | 70k |
| Train img balance | 3967 | 3967 | 14k - 56k | 14k - 56k | 3.53 - 14.11 | 3.53 - 14.11 | 7934 | 70k |
| Test default | 406 | 406 | 7k | 7k | 17.24 | 17.24 | 812 | 14k |
| Test id balance | 0 - 406 | 0 - 406 | 0 - 7k | 0 - 7k | 17.24 | 17.24 | 406 | 7k |
| Test img balance | 406 | 406 | 1.4k - 5.6k | 1.4k - 5.6k | 3.45 - 13.80 | 3.45 - 13.80 | 812 | 7k |

## A.4 MODEL VS. HUMAN SCATTERPLOTS

Figure 7 shows two example scatterplots comparing model L2 ratio (our proxy for confidence defined in section 7.2) against human accuracy on each question in the InterRace identification dataset (Dooley et al., 2021).

## A.5 RESULTS FOR MODELS TRAINED WITHOUT CLASS-BALANCED SAMPLING.

To explore the effect of class-balanced sampling on the results of our experiments, we train additional models without any oversampling strategies. Figures 8 - 10 show results of our experiments for MobileFaceNet and ResNet-152 models trained without oversampling. We find that most trends are

---

[3]https://github.com/ZhaoJ9014/face.evoLVe

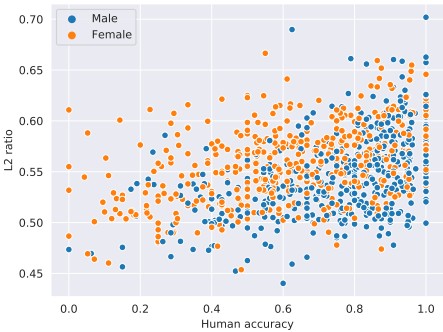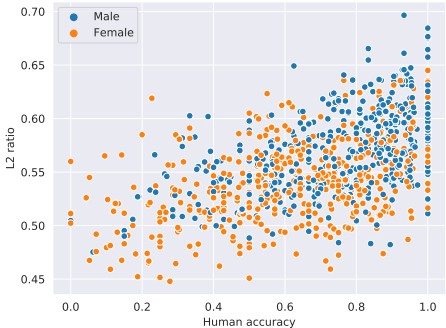

Figure 7: Scatterplots of model L2 ratio vs. human accuracy on each question in the InterRace identification dataset. Both models are MobileFaceNets trained with CosFace loss. (Left) a model trained on exclusively female images. (Right) a model trained on exclusively male images.

similar to ones observed in the models trained with class-balanced sampling, however models trained without oversampling are more robust to balance in the number of images per identity, see Figure 8. In particular, the effect of balancing the number of images (dashed lines) is similar to the effect of balancing the number of identities (solid lines) for all models, but ResNet-152 trained with ArcFace head. This leads us to a conclusion that using class-balanced sampling strategy is not beneficial in scenarios of severe imbalance in number of images per identity in face recognition models.

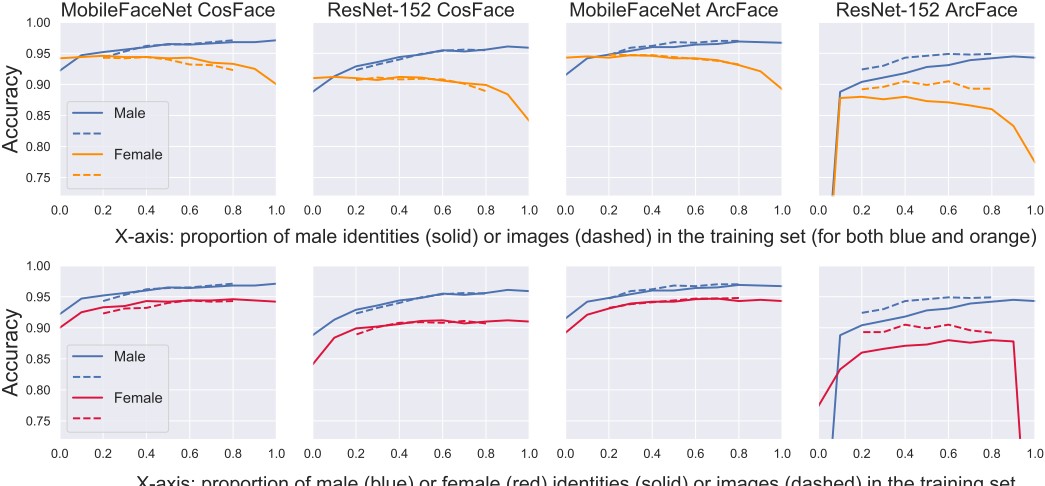

Figure 8: **Train Set Imbalance** Results of experiments that change the train set gender presentation balance for MobileFaceNet and ResNet-152 models trained **without class-balanced sampling**. Top row: male and female accuracy are plotted against the proportion of male data in the train set. Bottom row: for an alternate view, female accuracy is flipped horizontally, so that it is plotted against the proportion of *female* data in the train set. All models are evaluated on the default balanced test set.

## A.6 ADDITIONAL PLOTS AND TABLES

Figures 11 - 14 supplement those in sections 5 - 6. Figure 11 shows the results of the train set imbalance experiment when evaluated on the InterRace test set. Figures 12 - 14 show results for ResNet-50 (with ResNet-152 results shown again for comparison). Tables 2 - 7 precisely detail the number of male and female identities and images used in each experiment, as well as the accuracy on male and female targets and the female-to-male error ratio.

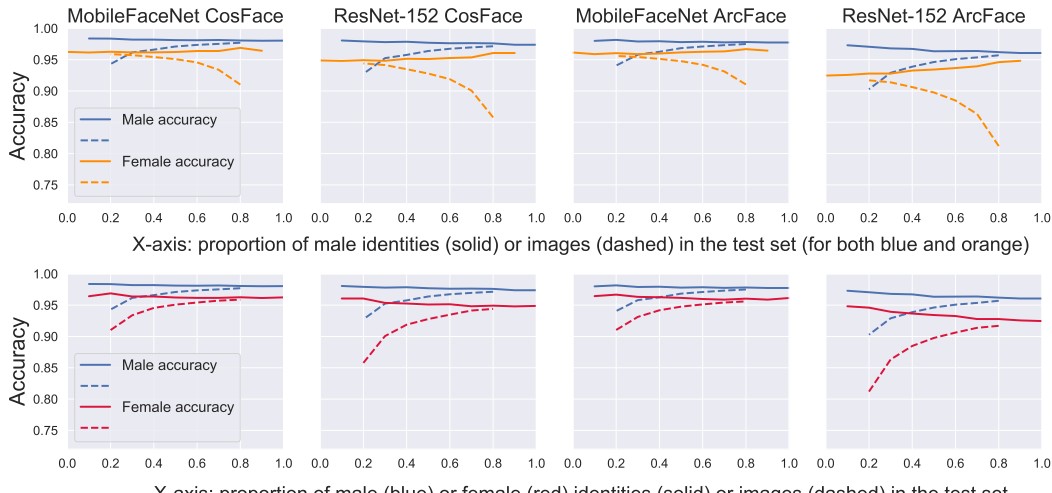

Figure 9: **Test Set Imbalance.** Results of experiments that change the test set gender presentation balance for MobileFaceNet and ResNet-152 models trained **without class-balanced sampling**. Top row: male and female accuracy are plotted against the proportion of male data in the test set. Bottom row: for an alternate view, female accuracy is flipped horizontally, so that it is plotted against the proportion of *female* data in the test set. All models are trained on the default balanced train set. For each experiment, the test set was split with 5 random seeds, and the results are averaged across seeds.

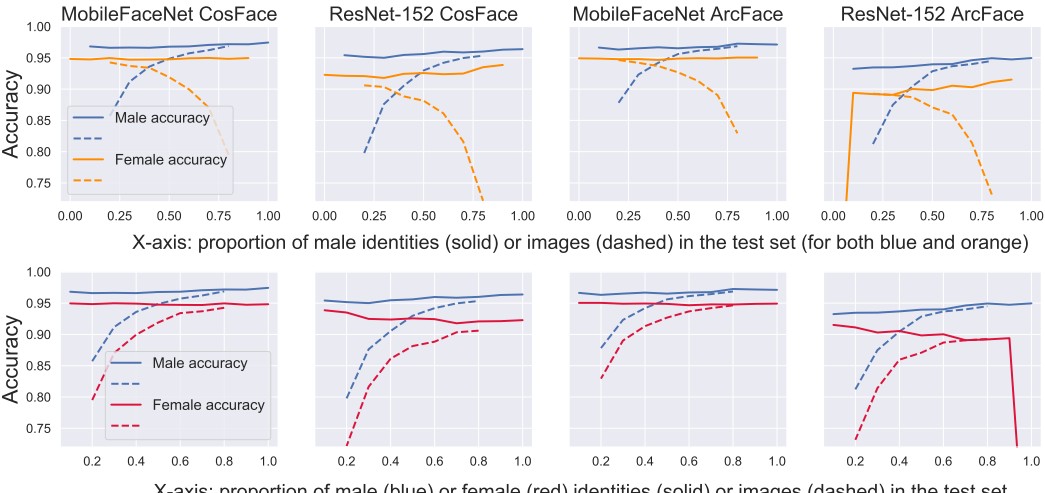

Figure 10: **Train & Test Set Imbalance.** Results of experiments that adjust the gender presentation balance in both the train and test set for MobileFaceNet and ResNet-152 models trained **without class-balanced sampling**. Top row: male and female accuracy are plotted against the proportion of male data used in both the train and test set. Bottom row: for an alternate view, female accuracy is flipped horizontally, so that it is plotted against the proportion of *female* data in both the train and test set. For each experiment, the test set was split with 5 random seeds, and the results are averaged across seeds.

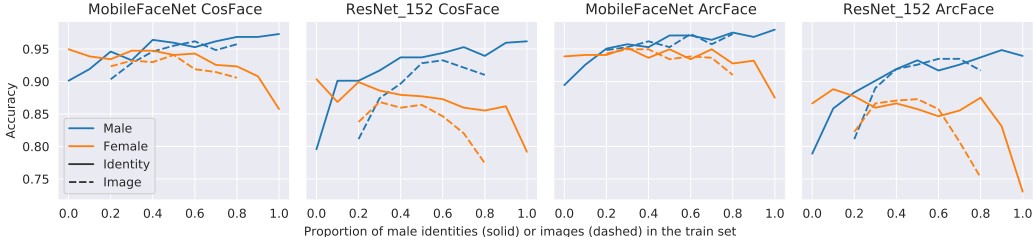

Figure 11: **Train Set Imbalance.** Results of experiments testing models trained with different gender presentation balance on the InterRace dataset. These plots are analogous to the first row of Figures 2 and 12.

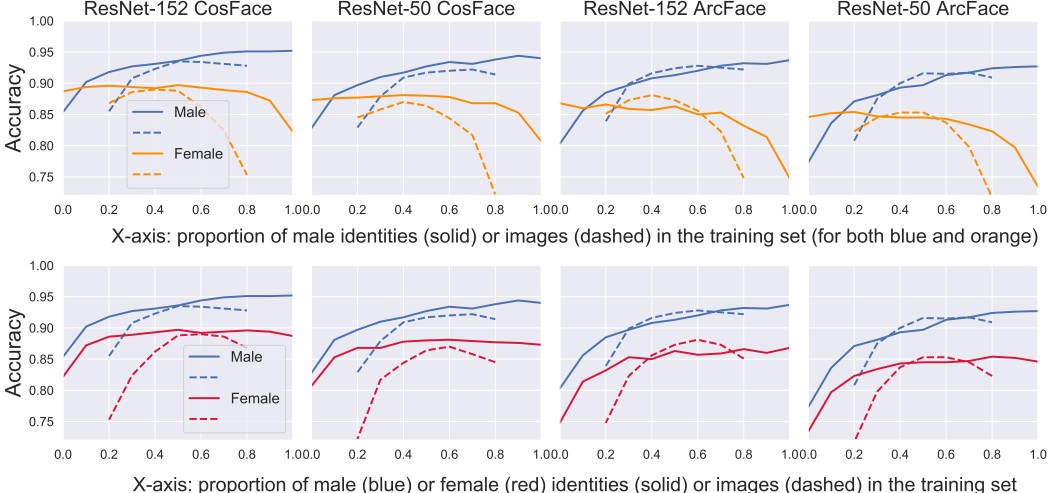

Figure 12: **Train Set Imbalance.** Results of experiments that change the train set gender presentation balance for ResNet-152 and ResNet-50 models. Top row: male and female accuracy are plotted against the proportion of male data in the train set. Bottom row: for an alternate view, female accuracy is flipped horizontally, so that it is plotted against the proportion of *female* data in the train set. All models are evaluated on the default balanced test set. Cf. Figure 2.

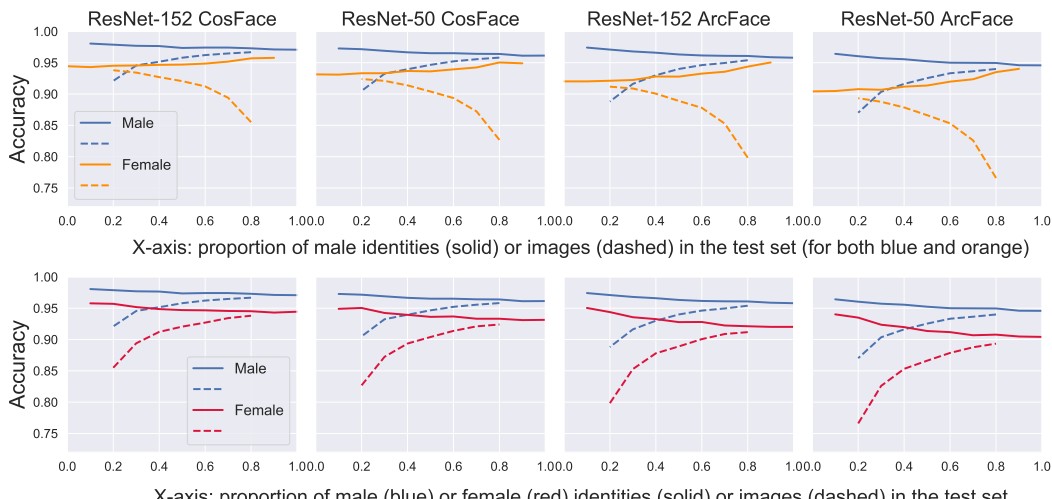

Figure 13: **Test Set Imbalance.** Results of experiments that change the test set gender presentation balance for ResNet-152 and ResNet-50 models. Top row: male and female accuracy are plotted against the proportion of male data in the test set. Bottom row: for an alternate view, female accuracy is flipped horizontally, so that it is plotted against the proportion of *female* data in the test set. All models are trained on the default balanced train set. For each experiment, the test set was split with 5 random seeds, and the results are averaged across seeds. Cf. Figure 3.

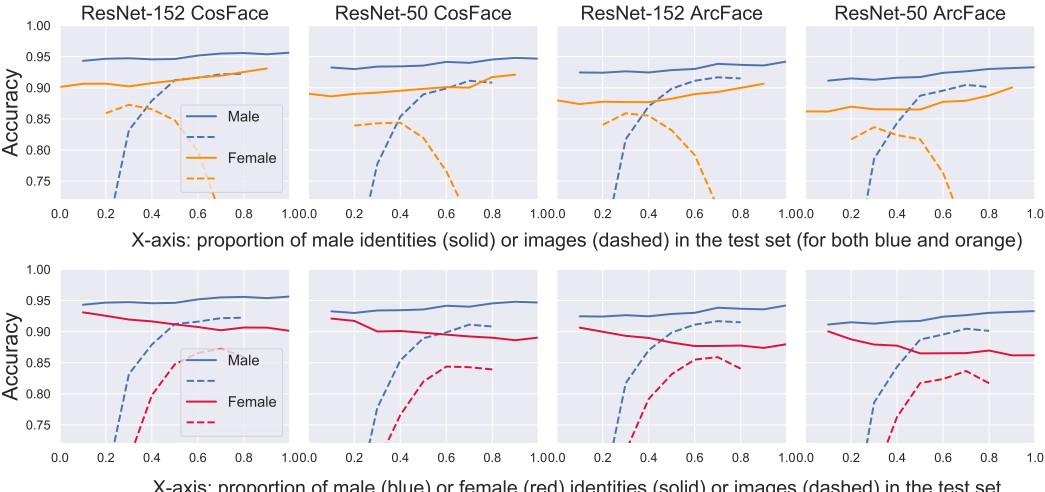

Figure 14: **Train & Test Set Imbalance.** Results of experiments that adjust the gender presentation balance in both the train and test set for ResNet-152 and ResNet-50 models. Top row: male and female accuracy are plotted against the proportion of male data used in both the train and test set. Bottom row: for an alternate view, female accuracy is flipped horizontally, so that it is plotted against the proportion of *female* data in both the train and test set. For each experiment, the test set was split with 5 random seeds, and the results are averaged across seeds. Cf. Figure 4.

Table 2: **Train Set Id Imbalance.** The female and male accuracy computed over the default balanced test set for models trained on data with various ratios of number of male and female identities. See details of the experiment in Section 4.1

| Model | Ids Ratio | M ids | F ids | M imgs | F imgs | M acc | F acc | Error Ratio |
|---|---|---|---|---|---|---|---|---|
| | 0 : 10 | 0 | 3967 | 0 | 70k | 0.918 | 0.938 | 0.76 |
| | 1 : 9 | 397 | 3570 | 7k | 63k | 0.941 | 0.939 | 1.03 |
| | 2 : 8 | 793 | 3174 | 14k | 56k | 0.946 | 0.941 | 1.09 |
| | 3 : 7 | 1190 | 2777 | 21k | 49k | 0.952 | 0.942 | 1.21 |
| | 4 : 6 | 1587 | 2380 | 28k | 42k | 0.958 | 0.940 | 1.43 |
| MFN CosFace | 5 : 5 | 1984 | 1984 | 35k | 35k | 0.961 | 0.940 | 1.54 |
| | 6 : 4 | 2380 | 1587 | 42k | 28k | 0.964 | 0.936 | 1.78 |
| | 7 : 3 | 2777 | 1190 | 49k | 21k | 0.965 | 0.935 | 1.86 |
| | 8 : 2 | 3174 | 793 | 56k | 14k | 0.964 | 0.928 | 2.00 |
| | 9 : 1 | 3570 | 397 | 63k | 7k | 0.968 | 0.924 | 2.37 |
| | 10 : 0 | 3967 | 0 | 70k | 0 | 0.968 | 0.887 | 3.53 |
| | 0 : 10 | 0 | 3967 | 0 | 70k | 0.911 | 0.937 | 0.71 |
| | 1 : 9 | 397 | 3570 | 7k | 63k | 0.937 | 0.940 | 0.95 |
| | 2 : 8 | 793 | 3174 | 14k | 56k | 0.948 | 0.939 | 1.17 |
| | 3 : 7 | 1190 | 2777 | 21k | 49k | 0.952 | 0.939 | 1.27 |
| | 4 : 6 | 1587 | 2380 | 28k | 42k | 0.953 | 0.941 | 1.26 |
| MFN ArcFace | 5 : 5 | 1984 | 1984 | 35k | 35k | 0.958 | 0.937 | 1.50 |
| | 6 : 4 | 2380 | 1587 | 42k | 28k | 0.965 | 0.937 | 1.80 |
| | 7 : 3 | 2777 | 1190 | 49k | 21k | 0.963 | 0.934 | 1.78 |
| | 8 : 2 | 3174 | 793 | 56k | 14k | 0.966 | 0.925 | 2.21 |
| | 9 : 1 | 3570 | 397 | 63k | 7k | 0.966 | 0.914 | 2.53 |
| | 10 : 0 | 3967 | 0 | 70k | 0 | 0.966 | 0.886 | 3.35 |
| | 0 : 10 | 0 | 3967 | 0 | 70k | 0.854 | 0.887 | 0.77 |
| | 1 : 9 | 397 | 3570 | 7k | 63k | 0.902 | 0.894 | 1.08 |
| | 2 : 8 | 793 | 3174 | 14k | 56k | 0.918 | 0.896 | 1.27 |
| | 3 : 7 | 1190 | 2777 | 21k | 49k | 0.927 | 0.894 | 1.45 |
| | 4 : 6 | 1587 | 2380 | 28k | 42k | 0.931 | 0.892 | 1.57 |
| ResNet-152 CosFace | 5 : 5 | 1984 | 1984 | 35k | 35k | 0.936 | 0.897 | 1.61 |
| | 6 : 4 | 2380 | 1587 | 42k | 28k | 0.944 | 0.893 | 1.91 |
| | 7 : 3 | 2777 | 1190 | 49k | 21k | 0.949 | 0.889 | 2.18 |
| | 8 : 2 | 3174 | 793 | 56k | 14k | 0.951 | 0.886 | 2.33 |
| | 9 : 1 | 3570 | 397 | 63k | 7k | 0.951 | 0.872 | 2.61 |
| | 10 : 0 | 3967 | 0 | 70k | 0 | 0.952 | 0.822 | 3.71 |
| | 0 : 10 | 0 | 3967 | 0 | 70k | 0.803 | 0.868 | 0.67 |
| | 1 : 9 | 397 | 3570 | 7k | 63k | 0.856 | 0.860 | 0.97 |
| | 2 : 8 | 793 | 3174 | 14k | 56k | 0.885 | 0.866 | 1.17 |
| | 3 : 7 | 1190 | 2777 | 21k | 49k | 0.897 | 0.859 | 1.37 |
| | 4 : 6 | 1587 | 2380 | 28k | 42k | 0.908 | 0.857 | 1.55 |
| ResNet-152 ArcFace | 5 : 5 | 1984 | 1984 | 35k | 35k | 0.913 | 0.863 | 1.57 |
| | 6 : 4 | 2380 | 1587 | 42k | 28k | 0.920 | 0.850 | 1.88 |
| | 7 : 3 | 2777 | 1190 | 49k | 21k | 0.928 | 0.853 | 2.04 |
| | 8 : 2 | 3174 | 793 | 56k | 14k | 0.932 | 0.832 | 2.47 |
| | 9 : 1 | 3570 | 397 | 63k | 7k | 0.931 | 0.814 | 2.70 |
| | 10 : 0 | 3967 | 0 | 70k | 0 | 0.937 | 0.748 | 4.00 |
| | 0 : 10 | 0 | 3967 | 0 | 70 | 0.828 | 0.873 | 0.74 |
| | 1 : 9 | 397 | 3570 | 7 | 63 | 0.881 | 0.876 | 1.04 |
| | 2 : 8 | 793 | 3174 | 14 | 56 | 0.897 | 0.877 | 1.19 |
| | 3 : 7 | 1190 | 2777 | 21 | 49 | 0.910 | 0.879 | 1.34 |
| | 4 : 6 | 1587 | 2380 | 28 | 42 | 0.917 | 0.881 | 1.43 |
| ResNet-50 CosFace | 5 : 5 | 1984 | 1984 | 35 | 35 | 0.927 | 0.880 | 1.64 |
| | 6 : 4 | 2380 | 1587 | 42 | 28 | 0.934 | 0.878 | 1.85 |
| | 7 : 3 | 2777 | 1190 | 49 | 21 | 0.931 | 0.868 | 1.91 |
| | 8 : 2 | 3174 | 793 | 56 | 14 | 0.938 | 0.868 | 2.13 |
| | 1 : 9 | 3570 | 397 | 63 | 7 | 0.944 | 0.853 | 2.63 |
| | 0 : 10 | 3967 | 0 | 70 | 0 | 0.940 | 0.807 | 3.22 |
| | 0 : 10 | 0 | 3967 | 0 | 70 | 0.773 | 0.846 | 0.68 |
| | 1 : 9 | 397 | 3570 | 7 | 63 | 0.836 | 0.852 | 0.90 |
| | 2 : 8 | 793 | 3174 | 14 | 56 | 0.871 | 0.854 | 1.13 |
| | 3 : 7 | 1190 | 2777 | 21 | 49 | 0.881 | 0.847 | 1.29 |
| | 4 : 6 | 1587 | 2380 | 28 | 42 | 0.893 | 0.845 | 1.45 |
| ResNet-50 ArcFace | 5 : 5 | 1984 | 1984 | 35 | 35 | 0.897 | 0.845 | 1.50 |
| | 6 : 4 | 2380 | 1587 | 42 | 28 | 0.913 | 0.843 | 1.80 |
| | 7 : 3 | 2777 | 1190 | 49 | 21 | 0.917 | 0.834 | 2.00 |
| | 8 : 2 | 3174 | 793 | 56 | 14 | 0.924 | 0.823 | 2.33 |
| | 1 : 9 | 3570 | 397 | 63 | 7 | 0.926 | 0.797 | 2.74 |
| | 0 : 10 | 3967 | 0 | 70 | 0 | 0.927 | 0.734 | 3.64 |

Table 3: **Train Set Img Imbalance.** The female and male accuracy computed over the default balanced test set for models trained on data with various ratios of number of images per male and female identity. See details of the experiment in Section 4.2

| Model | Img Ratio | # M ids | # F ids | # M imgs | # F imgs | M Acc | F Acc | Error Ratio |
|---|---|---|---|---|---|---|---|---|
| MFN CosFace | 2 : 8 | 3967 | 3967 | 14k | 56k | 0.932 | 0.927 | 1.07 |
| | 3 : 7 | 3967 | 3967 | 21k | 49k | 0.949 | 0.931 | 1.35 |
| | 4 : 6 | 3967 | 3967 | 28k | 42k | 0.955 | 0.931 | 1.53 |
| | 5 : 5 | 3967 | 3967 | 35k | 35k | 0.956 | 0.930 | 1.59 |
| | 6 : 4 | 3967 | 3967 | 42k | 28k | 0.959 | 0.929 | 1.73 |
| | 7 : 3 | 3967 | 3967 | 49k | 21k | 0.957 | 0.918 | 1.91 |
| | 8 : 2 | 3967 | 3967 | 56k | 14k | 0.957 | 0.892 | 2.51 |
| MFN ArcFace | 2 : 8 | 3967 | 3967 | 14k | 56k | 0.944 | 0.937 | 1.13 |
| | 3 : 7 | 3967 | 3967 | 21k | 49k | 0.953 | 0.939 | 1.30 |
| | 4 : 6 | 3967 | 3967 | 28k | 42k | 0.962 | 0.940 | 1.58 |
| | 5 : 5 | 3967 | 3967 | 35k | 35k | 0.962 | 0.939 | 1.61 |
| | 6 : 4 | 3967 | 3967 | 42k | 28k | 0.963 | 0.937 | 1.70 |
| | 7 : 3 | 3967 | 3967 | 49k | 21k | 0.961 | 0.929 | 1.82 |
| | 8 : 2 | 3967 | 3967 | 56k | 14k | 0.960 | 0.914 | 2.15 |
| ResNet-152 CosFace | 2 : 8 | 3967 | 3967 | 14k | 56k | 0.855 | 0.868 | 0.91 |
| | 3 : 7 | 3967 | 3967 | 21k | 49k | 0.908 | 0.886 | 1.24 |
| | 4 : 6 | 3967 | 3967 | 28k | 42k | 0.923 | 0.890 | 1.43 |
| | 5 : 5 | 3967 | 3967 | 35k | 35k | 0.935 | 0.888 | 1.72 |
| | 6 : 4 | 3967 | 3967 | 42k | 28k | 0.934 | 0.862 | 2.09 |
| | 7 : 3 | 3967 | 3967 | 49k | 21k | 0.931 | 0.824 | 2.55 |
| | 8 : 2 | 3967 | 3967 | 56k | 14k | 0.928 | 0.753 | 3.43 |
| ResNet-152 ArcFace | 2 : 8 | 3967 | 3967 | 14k | 56k | 0.839 | 0.851 | 0.93 |
| | 3 : 7 | 3967 | 3967 | 21k | 49k | 0.899 | 0.873 | 1.26 |
| | 4 : 6 | 3967 | 3967 | 28k | 42k | 0.916 | 0.881 | 1.42 |
| | 5 : 5 | 3967 | 3967 | 35k | 35k | 0.924 | 0.873 | 1.67 |
| | 6 : 4 | 3967 | 3967 | 42k | 28k | 0.928 | 0.856 | 2.00 |
| | 7 : 3 | 3967 | 3967 | 49k | 21k | 0.925 | 0.823 | 2.36 |
| | 8 : 2 | 3967 | 3967 | 56k | 14k | 0.922 | 0.748 | 3.23 |
| ResNet-50 CosFace | 2 : 8 | 3967 | 3967 | 14k | 56k | 0.829 | 0.845 | 0.91 |
| | 3 : 7 | 3967 | 3967 | 21k | 49k | 0.879 | 0.858 | 1.17 |
| | 4 : 6 | 3967 | 3967 | 28k | 42k | 0.909 | 0.870 | 1.43 |
| | 5 : 5 | 3967 | 3967 | 35k | 35k | 0.917 | 0.864 | 1.64 |
| | 6 : 4 | 3967 | 3967 | 42k | 28k | 0.920 | 0.844 | 1.95 |
| | 7 : 3 | 3967 | 3967 | 49k | 21k | 0.922 | 0.817 | 2.35 |
| | 8 : 2 | 3967 | 3967 | 56k | 14k | 0.914 | 0.722 | 3.23 |
| ResNet-50 ArcFace | 2 : 8 | 3967 | 3967 | 14k | 56k | 0.808 | 0.823 | 0.92 |
| | 3 : 7 | 3967 | 3967 | 21k | 49k | 0.875 | 0.845 | 1.24 |
| | 4 : 6 | 3967 | 3967 | 28k | 42k | 0.900 | 0.853 | 1.47 |
| | 5 : 5 | 3967 | 3967 | 35k | 35k | 0.916 | 0.853 | 1.75 |
| | 6 : 4 | 3967 | 3967 | 42k | 28k | 0.915 | 0.837 | 1.92 |
| | 7 : 3 | 3967 | 3967 | 49k | 21k | 0.917 | 0.798 | 2.43 |
| | 8 : 2 | 3967 | 3967 | 56k | 14k | 0.909 | 0.717 | 3.11 |

Table 4: **Test Set Id Imbalance.** The female and male accuracy for models trained on default train set computed on test set with various ratios of number of male and female identities. See details of experiment in Section 5.1.

| Model | Ids Ratio | # M ids | # F ids | # M imgs | # F imgs | M Acc | F Acc | Error Ratio |
|---|---|---|---|---|---|---|---|---|
| | 0 : 10 | 0 | 406 | 0 | 7000 | - | 0.961 | - |
| | 1 : 9 | 41 | 365 | 700 | 6300 | 0.983 | 0.961 | 2.25 |
| | 2 : 8 | 81 | 325 | 1400 | 5600 | 0.981 | 0.960 | 2.04 |
| | 3 : 7 | 122 | 284 | 2100 | 4900 | 0.981 | 0.960 | 2.09 |
| | 4 : 6 | 162 | 244 | 2800 | 4200 | 0.981 | 0.962 | 2.00 |
| MFN CosFace | 5 : 5 | 203 | 203 | 3500 | 3500 | 0.980 | 0.961 | 1.95 |
| | 6 : 4 | 244 | 162 | 4200 | 2800 | 0.980 | 0.963 | 1.83 |
| | 7 : 3 | 284 | 122 | 4900 | 2100 | 0.979 | 0.964 | 1.77 |
| | 8 : 2 | 325 | 81 | 5600 | 1400 | 0.979 | 0.969 | 1.45 |
| | 1 : 9 | 365 | 41 | 6300 | 700 | 0.978 | 0.962 | 1.72 |
| | 0 : 10 | 406 | 0 | 7000 | 0 | 0.978 | - | - |
| | 0 : 10 | 0 | 406 | 0 | 7000 | - | 0.959 | - |
| | 1 : 9 | 41 | 365 | 700 | 6300 | 0.980 | 0.959 | 2.07 |
| | 2 : 8 | 81 | 325 | 1400 | 5600 | 0.980 | 0.960 | 1.98 |
| | 3 : 7 | 122 | 284 | 2100 | 4900 | 0.981 | 0.958 | 2.17 |
| | 4 : 6 | 162 | 244 | 2800 | 4200 | 0.981 | 0.960 | 2.05 |
| MFN ArcFace | 5 : 5 | 203 | 203 | 3500 | 3500 | 0.979 | 0.961 | 1.89 |
| | 6 : 4 | 244 | 162 | 4200 | 2800 | 0.979 | 0.963 | 1.81 |
| | 7 : 3 | 284 | 122 | 4900 | 2100 | 0.979 | 0.962 | 1.84 |
| | 8 : 2 | 325 | 81 | 5600 | 1400 | 0.979 | 0.968 | 1.50 |
| | 1 : 9 | 365 | 41 | 6300 | 700 | 0.977 | 0.963 | 1.58 |
| | 0 : 10 | 406 | 0 | 7000 | 0 | 0.978 | - | - |
| | 0 : 10 | 0 | 406 | 0 | 7000 | - | 0.944 | - |
| | 1 : 9 | 41 | 365 | 700 | 6300 | 0.981 | 0.943 | 2.94 |
| | 2 : 8 | 81 | 325 | 1400 | 5600 | 0.979 | 0.945 | 2.58 |
| | 3 : 7 | 122 | 284 | 2100 | 4900 | 0.977 | 0.946 | 2.37 |
| | 4 : 6 | 162 | 244 | 2800 | 4200 | 0.977 | 0.947 | 2.28 |
| ResNet-152 CosFace | 5 : 5 | 203 | 203 | 3500 | 3500 | 0.974 | 0.947 | 2.01 |
| | 6 : 4 | 244 | 162 | 4200 | 2800 | 0.974 | 0.949 | 1.99 |
| | 7 : 3 | 284 | 122 | 4900 | 2100 | 0.974 | 0.952 | 1.87 |
| | 8 : 2 | 325 | 81 | 5600 | 1400 | 0.973 | 0.957 | 1.59 |
| | 1 : 9 | 365 | 41 | 6300 | 700 | 0.971 | 0.958 | 1.47 |
| | 0 : 10 | 406 | 0 | 7000 | 0 | 0.971 | - | - |
| | 0 : 10 | 0 | 406 | 0 | 7000 | - | 0.920 | - |
| | 1 : 9 | 41 | 365 | 700 | 6300 | 0.974 | 0.920 | 3.09 |
| | 2 : 8 | 81 | 325 | 1400 | 5600 | 0.971 | 0.921 | 2.72 |
| | 3 : 7 | 122 | 284 | 2100 | 4900 | 0.968 | 0.922 | 2.42 |
| | 4 : 6 | 162 | 244 | 2800 | 4200 | 0.966 | 0.928 | 2.12 |
| ResNet-152 ArcFace | 5 : 5 | 203 | 203 | 3500 | 3500 | 0.963 | 0.928 | 1.96 |
| | 6 : 4 | 244 | 162 | 4200 | 2800 | 0.962 | 0.933 | 1.76 |
| | 7 : 3 | 284 | 122 | 4900 | 2100 | 0.961 | 0.936 | 1.65 |
| | 8 : 2 | 325 | 81 | 5600 | 1400 | 0.961 | 0.944 | 1.43 |
| | 1 : 9 | 365 | 41 | 6300 | 700 | 0.959 | 0.950 | 1.20 |
| | 0 : 10 | 406 | 0 | 7000 | 0 | 0.958 | - | - |
| | 0 : 10 | 0 | 406 | 0 | 7000 | - | 0.931 | - |
| | 1 : 9 | 41 | 365 | 700 | 6300 | 0.973 | 0.931 | 2.54 |
| | 2 : 8 | 81 | 325 | 1400 | 5600 | 0.972 | 0.933 | 2.35 |
| | 3 : 7 | 122 | 284 | 2100 | 4900 | 0.969 | 0.933 | 2.15 |
| | 4 : 6 | 162 | 244 | 2800 | 4200 | 0.967 | 0.937 | 1.89 |
| ResNet-50 CosFace | 5 : 5 | 203 | 203 | 3500 | 3500 | 0.965 | 0.936 | 1.83 |
| | 6 : 4 | 244 | 162 | 4200 | 2800 | 0.965 | 0.939 | 1.74 |
| | 7 : 3 | 284 | 122 | 4900 | 2100 | 0.964 | 0.942 | 1.61 |
| | 8 : 2 | 325 | 81 | 5600 | 1400 | 0.964 | 0.950 | 1.38 |
| | 1 : 9 | 365 | 41 | 6300 | 700 | 0.961 | 0.949 | 1.31 |
| | 0 : 10 | 406 | 0 | 7000 | 0 | 0.961 | - | - |
| | 0 : 10 | 0 | 406 | 0 | 7000 | - | 0.904 | - |
| | 1 : 9 | 41 | 365 | 700 | 6300 | 0.964 | 0.905 | 2.66 |
| | 2 : 8 | 81 | 325 | 1400 | 5600 | 0.960 | 0.908 | 2.33 |
| | 3 : 7 | 122 | 284 | 2100 | 4900 | 0.957 | 0.907 | 2.18 |
| | 4 : 6 | 162 | 244 | 2800 | 4200 | 0.956 | 0.912 | 1.99 |
| ResNet-50 ArcFace | 5 : 5 | 203 | 203 | 3500 | 3500 | 0.952 | 0.914 | 1.82 |
| | 6 : 4 | 244 | 162 | 4200 | 2800 | 0.950 | 0.920 | 1.60 |
| | 7 : 3 | 284 | 122 | 4900 | 2100 | 0.950 | 0.924 | 1.52 |
| | 8 : 2 | 325 | 81 | 5600 | 1400 | 0.950 | 0.935 | 1.29 |
| | 1 : 9 | 365 | 41 | 6300 | 700 | 0.946 | 0.940 | 1.11 |
| | 0 : 10 | 406 | 0 | 7000 | 0 | 0.946 | - | - |

Table 5: **Test Set Img Imbalance.** The female and male accuracy for models trained on default train set computed on test set with various ratios of number of images per male and female identities. See details of the experiment in Section 5.2

| Model | Img Ratio | # M ids | # F ids | # M imgs | # F imgs | M Acc | F Acc | Error Ratio |
|---|---|---|---|---|---|---|---|---|
| MFN CosFace | 2 : 8 | 406 | 406 | 1400 | 5600 | 0.941 | 0.957 | 0.72 |
| | 3 : 7 | 406 | 406 | 2100 | 4900 | 0.959 | 0.956 | 1.06 |
| | 4 : 6 | 406 | 406 | 2800 | 4200 | 0.962 | 0.952 | 1.27 |
| | 5 : 5 | 406 | 406 | 3500 | 3500 | 0.967 | 0.946 | 1.64 |
| | 6 : 4 | 406 | 406 | 4200 | 2800 | 0.970 | 0.940 | 2.01 |
| | 7 : 3 | 406 | 406 | 4900 | 2100 | 0.973 | 0.925 | 2.75 |
| | 8 : 2 | 406 | 406 | 5600 | 1400 | 0.975 | 0.894 | 4.23 |
| MFN ArcFace | 2 : 8 | 406 | 406 | 1400 | 5600 | 0.939 | 0.956 | 0.72 |
| | 3 : 7 | 406 | 406 | 2100 | 4900 | 0.956 | 0.954 | 1.03 |
| | 4 : 6 | 406 | 406 | 2800 | 4200 | 0.961 | 0.951 | 1.26 |
| | 5 : 5 | 406 | 406 | 3500 | 3500 | 0.966 | 0.947 | 1.54 |
| | 6 : 4 | 406 | 406 | 4200 | 2800 | 0.969 | 0.941 | 1.91 |
| | 7 : 3 | 406 | 406 | 4900 | 2100 | 0.972 | 0.928 | 2.58 |
| | 8 : 2 | 406 | 406 | 5600 | 1400 | 0.974 | 0.901 | 3.87 |
| ResNet-152 CosFace | 2 : 8 | 406 | 406 | 1400 | 5600 | 0.921 | 0.938 | 0.78 |
| | 3 : 7 | 406 | 406 | 2100 | 4900 | 0.946 | 0.934 | 1.21 |
| | 4 : 6 | 406 | 406 | 2800 | 4200 | 0.952 | 0.927 | 1.51 |
| | 5 : 5 | 406 | 406 | 3500 | 3500 | 0.958 | 0.921 | 1.89 |
| | 6 : 4 | 406 | 406 | 4200 | 2800 | 0.962 | 0.912 | 2.32 |
| | 7 : 3 | 406 | 406 | 4900 | 2100 | 0.965 | 0.894 | 3.01 |
| | 8 : 2 | 406 | 406 | 5600 | 1400 | 0.967 | 0.855 | 4.37 |
| ResNet-152 ArcFace | 2 : 8 | 406 | 406 | 1400 | 5600 | 0.888 | 0.912 | 0.79 |
| | 3 : 7 | 406 | 406 | 2100 | 4900 | 0.916 | 0.909 | 1.09 |
| | 4 : 6 | 406 | 406 | 2800 | 4200 | 0.930 | 0.901 | 1.42 |
| | 5 : 5 | 406 | 406 | 3500 | 3500 | 0.940 | 0.889 | 1.85 |
| | 6 : 4 | 406 | 406 | 4200 | 2800 | 0.946 | 0.878 | 2.27 |
| | 7 : 3 | 406 | 406 | 4900 | 2100 | 0.950 | 0.853 | 2.92 |
| | 8 : 2 | 406 | 406 | 5600 | 1400 | 0.954 | 0.798 | 4.38 |
| ResNet-50 CosFace | 2 : 8 | 406 | 406 | 1400 | 5600 | 0.905 | 0.924 | 0.80 |
| | 3 : 7 | 406 | 406 | 2100 | 4900 | 0.932 | 0.921 | 1.17 |
| | 4 : 6 | 406 | 406 | 2800 | 4200 | 0.940 | 0.914 | 1.43 |
| | 5 : 5 | 406 | 406 | 3500 | 3500 | 0.947 | 0.904 | 1.80 |
| | 6 : 4 | 406 | 406 | 4200 | 2800 | 0.952 | 0.894 | 2.23 |
| | 7 : 3 | 406 | 406 | 4900 | 2100 | 0.956 | 0.872 | 2.87 |
| | 8 : 2 | 406 | 406 | 5600 | 1400 | 0.958 | 0.827 | 4.14 |
| ResNet-50 ArcFace | 2 : 8 | 406 | 406 | 1400 | 5600 | 0.870 | 0.893 | 0.82 |
| | 3 : 7 | 406 | 406 | 2100 | 4900 | 0.904 | 0.888 | 1.17 |
| | 4 : 6 | 406 | 406 | 2800 | 4200 | 0.916 | 0.879 | 1.45 |
| | 5 : 5 | 406 | 406 | 3500 | 3500 | 0.925 | 0.866 | 1.79 |
| | 6 : 4 | 406 | 406 | 4200 | 2800 | 0.933 | 0.853 | 2.20 |
| | 7 : 3 | 406 | 406 | 4900 | 2100 | 0.936 | 0.826 | 2.74 |
| | 8 : 2 | 406 | 406 | 5600 | 1400 | 0.940 | 0.766 | 3.90 |

Table 6: **Train & Test Set Id Imbalance.** The female and male accuracy for models trained and tested on data with the same ratios of male and female identities. See details of experiment in Section 6.

| Model | Ids Ratio | M Acc | F Acc | Error Ratio |
|---|---|---|---|---|
| MFN CosFace | 0 : 10 | - | 0.945 | - |
| | 1 : 9 | 0.963 | 0.943 | 1.54 |
| | 2 : 8 | 0.966 | 0.947 | 1.56 |
| | 3 : 7 | 0.964 | 0.943 | 1.57 |
| | 4 : 6 | 0.967 | 0.945 | 1.63 |
| | 5 : 5 | 0.965 | 0.943 | 1.63 |
| | 6 : 4 | 0.968 | 0.947 | 1.63 |
| | 7 : 3 | 0.968 | 0.946 | 1.66 |
| | 8 : 2 | 0.969 | 0.946 | 1.72 |
| | 1 : 9 | 0.971 | 0.951 | 1.68 |
| | 0 : 10 | 0.972 | - | - |
| MFN ArcFace | 0 : 10 | - | 0.945 | - |
| | 1 : 9 | 0.962 | 0.946 | 1.42 |
| | 2 : 8 | 0.962 | 0.947 | 1.42 |
| | 3 : 7 | 0.962 | 0.943 | 1.52 |
| | 4 : 6 | 0.961 | 0.945 | 1.41 |
| | 5 : 5 | 0.964 | 0.944 | 1.54 |
| | 6 : 4 | 0.968 | 0.944 | 1.72 |
| | 7 : 3 | 0.967 | 0.946 | 1.61 |
| | 8 : 2 | 0.969 | 0.947 | 1.71 |
| | 1 : 9 | 0.968 | 0.949 | 1.63 |
| | 0 : 10 | 0.969 | - | - |
| ResNet-152 CosFace | 0 : 10 | - | 0.901 | - |
| | 1 : 9 | 0.943 | 0.906 | 1.65 |
| | 2 : 8 | 0.947 | 0.907 | 1.75 |
| | 3 : 7 | 0.947 | 0.902 | 1.86 |
| | 4 : 6 | 0.946 | 0.907 | 1.70 |
| | 5 : 5 | 0.946 | 0.912 | 1.64 |
| | 6 : 4 | 0.952 | 0.916 | 1.73 |
| | 7 : 3 | 0.955 | 0.919 | 1.79 |
| | 8 : 2 | 0.956 | 0.925 | 1.69 |
| | 1 : 9 | 0.954 | 0.931 | 1.49 |
| | 0 : 10 | 0.956 | - | - |
| ResNet-152 ArcFace | 0 : 10 | - | 0.880 | - |
| | 1 : 9 | 0.925 | 0.874 | 1.67 |
| | 2 : 8 | 0.924 | 0.878 | 1.61 |
| | 3 : 7 | 0.926 | 0.877 | 1.67 |
| | 4 : 6 | 0.925 | 0.877 | 1.63 |
| | 5 : 5 | 0.928 | 0.882 | 1.64 |
| | 6 : 4 | 0.930 | 0.890 | 1.58 |
| | 7 : 3 | 0.938 | 0.893 | 1.73 |
| | 8 : 2 | 0.937 | 0.900 | 1.59 |
| | 1 : 9 | 0.936 | 0.906 | 1.46 |
| | 0 : 10 | 0.942 | - | - |
| ResNet-50 CosFace | 0 : 10 | - | 0.890 | - |
| | 1 : 9 | 0.933 | 0.886 | 1.69 |
| | 2 : 8 | 0.930 | 0.890 | 1.57 |
| | 3 : 7 | 0.934 | 0.892 | 1.63 |
| | 4 : 6 | 0.934 | 0.895 | 1.60 |
| | 5 : 5 | 0.936 | 0.898 | 1.58 |
| | 6 : 4 | 0.942 | 0.901 | 1.70 |
| | 7 : 3 | 0.940 | 0.900 | 1.66 |
| | 8 : 2 | 0.945 | 0.917 | 1.52 |
| | 1 : 9 | 0.948 | 0.921 | 1.52 |
| | 0 : 10 | 0.947 | - | - |
| ResNet-50 ArcFace | 0 : 10 | - | 0.862 | - |
| | 1 : 9 | 0.911 | 0.862 | 1.56 |
| | 2 : 8 | 0.915 | 0.870 | 1.53 |
| | 3 : 7 | 0.913 | 0.865 | 1.54 |
| | 4 : 6 | 0.916 | 0.865 | 1.61 |
| | 5 : 5 | 0.917 | 0.865 | 1.63 |
| | 6 : 4 | 0.924 | 0.877 | 1.61 |
| | 7 : 3 | 0.926 | 0.879 | 1.64 |
| | 8 : 2 | 0.930 | 0.888 | 1.61 |
| | 1 : 9 | 0.932 | 0.900 | 1.46 |
| | 0 : 10 | 0.933 | - | - |

Table 7: **Train & Test Set Img Imbalance.** The female and male accuracy for models trained and tested on data with the same ratios of number of images per male and female identity. See details of experiment in Section 6.

| Model | Img Ratio | M Acc | F Acc | Error Ratio |
|---|---|---|---|---|
| | 2 : 8 | 0.821 | 0.923 | 0.43 |
| | 3 : 7 | 0.901 | 0.922 | 0.78 |
| | 4 : 6 | 0.928 | 0.919 | 1.12 |
| MFN CosFace | 5 : 5 | 0.942 | 0.906 | 1.62 |
| | 6 : 4 | 0.952 | 0.892 | 2.24 |
| | 7 : 3 | 0.951 | 0.848 | 3.12 |
| | 8 : 2 | 0.954 | 0.740 | 5.70 |
| | 2 : 8 | 0.854 | 0.932 | 0.46 |
| | 3 : 7 | 0.916 | 0.933 | 0.79 |
| | 4 : 6 | 0.937 | 0.927 | 1.16 |
| MFN ArcFace | 5 : 5 | 0.951 | 0.919 | 1.64 |
| | 6 : 4 | 0.955 | 0.907 | 2.09 |
| | 7 : 3 | 0.957 | 0.871 | 3.01 |
| | 8 : 2 | 0.958 | 0.779 | 5.31 |
| | 2 : 8 | 0.657 | 0.859 | 0.41 |
| | 3 : 7 | 0.832 | 0.873 | 0.76 |
| | 4 : 6 | 0.879 | 0.866 | 1.11 |
| ResNet-152 CosFace | 5 : 5 | 0.912 | 0.848 | 1.74 |
| | 6 : 4 | 0.916 | 0.798 | 2.41 |
| | 7 : 3 | 0.922 | 0.695 | 3.90 |
| | 8 : 2 | 0.922 | 0.483 | 6.66 |
| | 2 : 8 | 0.638 | 0.840 | 0.44 |
| | 3 : 7 | 0.817 | 0.859 | 0.77 |
| | 4 : 6 | 0.870 | 0.855 | 1.12 |
| ResNet-152 ArcFace | 5 : 5 | 0.899 | 0.832 | 1.66 |
| | 6 : 4 | 0.911 | 0.792 | 2.34 |
| | 7 : 3 | 0.917 | 0.708 | 3.51 |
| | 8 : 2 | 0.915 | 0.488 | 6.02 |
| | 2 : 8 | 0.611 | 0.839 | 0.41 |
| | 3 : 7 | 0.777 | 0.843 | 0.71 |
| | 4 : 6 | 0.854 | 0.844 | 1.07 |
| ResNet-50 CosFace | 5 : 5 | 0.889 | 0.820 | 1.63 |
| | 6 : 4 | 0.899 | 0.766 | 2.31 |
| | 7 : 3 | 0.911 | 0.688 | 3.51 |
| | 8 : 2 | 0.908 | 0.449 | 6.00 |
| | 2 : 8 | 0.579 | 0.817 | 0.43 |
| | 3 : 7 | 0.786 | 0.837 | 0.76 |
| | 4 : 6 | 0.844 | 0.824 | 1.13 |
| ResNet-50 ArcFace | 5 : 5 | 0.887 | 0.817 | 1.62 |
| | 6 : 4 | 0.895 | 0.763 | 2.27 |
| | 7 : 3 | 0.905 | 0.666 | 3.50 |
| | 8 : 2 | 0.901 | 0.450 | 5.56 |

