# OpenReview forum: "A Deep Dive into Dataset Imbalance and Bias in Face Identification"
_ICLR.cc/2023/Conference — Submitted to ICLR 2023_

### Official Review · Reviewer_vAAf · 2022-10-25

**Confidence:** 4
**Correctness:** 3
**Technical Novelty And Significance:** 2
**Empirical Novelty And Significance:** 2
**Recommendation:** 3

**Clarity, Quality, Novelty And Reproducibility:**

Clarity: This paper is clear and easy to understand
Quality: This paper is of average quality and has some loopholes
Novelty: The direction of this paper is innovative
Reproducibility: This paper is reproducible

**Strength And Weaknesses:**

Strength:
1. The problem of data imbalance is a good direction and deserves further study.
2. The authors have carried out experiments on the problem of data imbalance in different situations and draw conclusions according to the experimental results.

Weaknesses:
1. The author carried out experiments (session 4 and session 5) respectively under the condition that the training set and the test set are not balanced, but the conclusion obtained is relatively shallow. At the same time, in the case of imbalanced training set images, the abnormal occurrence is caused by resampling, that is, the bias is partly caused by human factors and the result is not rigorous enough.
2. The authors conclude in session 7.1 that females are generally more difficult to identify than males, but the authors only conducted experiments on the Celeb A dataset.  There are differences in the difficulty of different data sets.  If the experiment can be carried out in more datasets, the conclusion will be more convincing.
3. In session 7.2, the author argues that there are reasons for bias in humans and machines are different. But the author does not analyze what reason is and why they are different. I think this conclusion is meaningless.

**Summary Of The Paper:**

This paper aims to explore the problem of data set imbalance in face identification systems. They specifically focus on imbalance with respect to gender presentation. Aiming at the imbalances(both in terms of identities or images per identity) in the train set and the test set, the author conducted several experiments on the Celeb A dataset and got some conclusions.

**Summary Of The Review:**

The authors have carried out various experiments, but the conclusion is relatively simple. The authors reveal many common errors in bias calculation in face recognition tasks, but do not provide effective strategies to avoid these problems

---

> ### Author Response · Authors · 2022-11-19
> **Response to Reviewer vAAf**
>
> Thank you for your thoughtful review and feedback, we address each of your comments below:
>
> 1. In our work, we study how different kinds of imbalance, such as imbalance with respect to the number of images and identities, in the train and in the gallery set affect bias in face identification systems. While train data imbalance is a relatively explored topic, gallery set imbalance has not been considered prior to our work to the best of our knowledge. Our findings show that each type of imbalance has a distinct effect on a model’s performance on each gender presentation. One of our observations is that overrepresenting the target demographic group in the train data can hurt that demographic group when an oversampling strategy is used to “balance” the skewed distribution of data. We additionally conduct experiments without oversampling strategy and show the results in Figure 8 in Appendix.
>
> 2. In Section 7.1 we test randomly initialized feature extractors on galleries with varying levels of image imbalance. We find that random models have higher male performance even when the gallery set is perfectly balanced, which suggests that there are sources of bias in face identification systems which lie outside of the data imbalance aspect. However we do not conclude that female images are generally more difficult to recognize than male images.
>
> 3. In Section 7.2 we compare human and model bias on InterRace dataset. We find that although both humans and models find identifying females more difficult than identifying males, particular images that are easier or harder to identify appear to differ between models and humans. We believe that this is an interesting observation showing that gender bias in face identification systems and humans come from different sources.

---

### Official Review · Reviewer_qe6A · 2022-10-25

**Confidence:** 4
**Correctness:** 3
**Technical Novelty And Significance:** 2
**Empirical Novelty And Significance:** 2
**Recommendation:** 3

**Clarity, Quality, Novelty And Reproducibility:**

The novelty is limited (see the summary).

Clarity is not good in terms of precisely defining the task and the protected attributes.

**Strength And Weaknesses:**

Strengths:

1. Fair number of experiments done on a standard dataset.
2. Multiple models investigated.
3. Extensive analysis conducted

Weaknesses:
1. Limited novelty in the sense that this is purely an analysis paper, and that too this type of analysis has been done before if not on the same dataset.
2. Paper not well-written, e.g., even the task isn't clearly described (see the summary).
3. No insights is provided into what exactly is the (likely) reason for the observations.
4. No attempt to alleviate the bias is made.



**Summary Of The Paper:**

The paper can be categorized into an "analysis type" paper, which doesn't propose anything new; instead, it demonstrates that standard image classification modeles, such as RestNet, MobileNet etc., can lead to biased predictions on face recognition tasks and datasets.


**Summary Of The Review:**

The paper can be categorized into an "analysis type" paper, which doesn't propose anything new; instead, it demonstrates that standard image classification modeles, such as RestNet, MobileNet etc., can lead to biased predictions on face recognition tasks and datasets.

The paper is poorly written. It's not even clear after reading Section 3 (Face Identification Setup) that what the task actually is. Is the model trained to predict a particular class (ot of say p ones), i.e., $\theta: \vec{x} \mapsto \mathbb{Z}_p$ or is the task that of metric learning, i.e., $\theta: \vec{x} \times \vec{z} \mapsto \mathbb{R}$?

In addition to gender, there're plenty of other attributes in the CelebA dataset that can be considered for stereotypical biases (checking for parity in the posteriors), e.g., "attractiveness" is one such feature, and there could be easily be biases discovered by a combination of more than one attribute, e.g., "females wearing glasses are less attractive" etc.

The paper reports the biases but doesn't propose a method to alleviate them, which in my opinion, severly limits its contributions in terms of novelty.

The paper lacks any kind of analysis. What's presented as actionable insights, such as "overrepresenting the target demographic group can sometimes hurt that group...", an academic paper should provide insights into why is that the case? What's the difference between the performance from different models? Is MobileNet more biased than ResNet? If so, why, if not then why not?

---

> ### Author Response · Authors · 2022-11-19
> **Response to Reviewer qe6A**
>
> Thank you for your thoughtful review and feedback, we address your comments below:
>
> 1. To the best of our knowledge, our paper is the first work exploring how different kinds of data imbalance, including imbalance with respect to the number of images and identities, in the train and gallery data affects bias of face identification systems. We would appreciate it if you could point us to previous works doing similar analysis. We would be happy to compare contributions and cite these related works.
>
> 2. Regarding the task, we conduct experiments in a face identification setup, where the goal is to match a probe image against a set of images (called the gallery) with known identities. Face identification models are trained as image classifiers, where classes correspond to identities. At inference time, they identify the closest gallery image to a probe image using distance measures in the feature space.
>
> 3. Regarding additional attributes and stereotypical biases, although CelebA indeed contains multiple attribute annotations, such as “attractiveness”, which could be used for exploring stereotypical biases, our work focuses on different aspects of model bias, such as the effect of different kinds of data imbalance on bias of face identification systems.
>
> 4. *No insights is provided into what exactly is the (likely) reason for the observations.*
> Although we do not elaborate on the reasons for certain trends in the concluding section, we discuss them in detail in the corresponding Sections 4,5,6,7. In particular, we discuss how different kinds of data imbalance in the train or gallery sets affect bias in face identification models and provide evidence through experiments. In the “Actionable Insights” section we summarize our findings for practitioners.

---

### Official Review · Reviewer_AVTo · 2022-11-02

**Confidence:** 2
**Correctness:** 3
**Technical Novelty And Significance:** 2
**Empirical Novelty And Significance:** 2
**Recommendation:** 5

**Clarity, Quality, Novelty And Reproducibility:**

The writing is clear and straightforward. But the technical contribution of this paper is limited. This paper is easy to reproduce the statistics reported in the paper.

**Details Of Ethics Concerns:**

No ethics concerns.

**Strength And Weaknesses:**

Strength:

Bias in face recognition is a timely topic.

This paper provided comprehensive statistics regarding bias in the training set and test set.

Weakness:

In Section 7.1, 'We observe that both models have higher male performance when the test set is perfectly balanced'. This observation is not explored in depth. Is it because female images usually contain make-up?

In Fig 6, there is no black line in the figure. The legends (identity, image)  are confusing. It is better to show male identity, male image and so on.

There is no reference to MobileFaceNet, CosFace and ArcFace.

The technical contribution of this paper is limited as only sampling strategy is used with some simple statistics on the results.

There are some useful conclusions in Section 8, but it is not very insightful for training unbiased face recognition models

**Summary Of The Paper:**

This paper unravelled the complex effects that dataset imbalance can have on the model bias for face identification systems.

This paper separately considered imbalance in terms of identities and images per identity in both the train set and the test set.

This paper thoroughly explored the effects of each kind of imbalance possible in face identification, and discuss factors which may impact bias.

**Summary Of The Review:**

Bias in face recognition is a timely topic. This paper employed data sampling techniques on different dimensions, such as training data, test data, identity and image. Extensive results supported some conclusions on the bias problem of face recognition. However, the technical contribution of this paper is limited and some observations are not in-depth.

---

> ### Author Response · Authors · 2022-11-19
> **Response to Reviewer AVTo**
>
> Thank you for your thoughtful review and feedback, we address your comments below:
>
> 1. It is a very interesting question why randomly initialized models have higher male performance when the test set is perfectly balanced. One reason could indeed be a larger variance in the input (image) space for female images, for example because of make-up or more variability in the hairstyle. However, it is not entirely clear how to measure the variance in the input space. We did experiments with measuring the variance in the feature space of a neural network and found that female images have higher within-class variance, than male images. However, we note that this could be a result of higher variance in the input space OR algorithmic bias of a neural network itself.
>
>
> 2. *In Fig 6, there is no black line in the figure.*
> We include the black solid and dashed lines in the legend of Fig 6 to distinguish between the balance with respect to the number of identities (solid lines) and balance with respect to the number of images (dashed line).
>
> 3. *There is no reference to MobileFaceNet, CosFace and ArcFace.*
> Thank you for pointing this out, we have now updated our draft to include the references.
>
> 4. Regarding our contribution, our paper provides insights into how different kinds of data imbalance affect bias of face identification systems and how train and gallery set imbalances may interact with each other and lead to underestimation of true bias. We hope that our findings will help practitioners avoid common mistakes in bias computations for many facial recognition tasks. We additionally analyze whether data imbalance captures all the inherent bias and how the bias we see in face identification models compares to human biases.

---

### Decision · Program_Chairs · 2023-01-20

**Decision:**

Reject

**Justification For Why Not Higher Score:**

N/A

**Justification For Why Not Lower Score:**

N/A

**Metareview: Summary, Strengths And Weaknesses:**

The paper studies face identification and demonstrates how various types of data imbalance can lead to model bias. AC agrees with the reviewers that albeit it’s an important problem, limited technical contribution (how to resolve existing model failures) and a narrow application domain (the paper studies face identification) are two critical issues that place the contributions below acceptance bar. We hope the reviews are useful to improve the manuscript.